# Clinical Application of the Association between Genetic Alteration and Intraoperative Fluorescence Activity of 5-Aminolevulinic Acid during the Resection of Brain Metastasis of Lung Adenocarcinoma

**DOI:** 10.3390/cancers16010088

**Published:** 2023-12-23

**Authors:** Hyeon Yeong Jeong, Won Jun Suh, Seung Hwan Kim, Taek Min Nam, Ji Hwan Jang, Kyu Hong Kim, Seok Hyun Kim, Young Zoon Kim

**Affiliations:** 1Division of Cerebrovascular Disease and Department of Neurosurgery, Samsung Changwon Hospital, Sungkyunkwan University of School of Medicine, Changwon 51353, Republic of Korea; wjdgusdud23@naver.com (H.Y.J.); aajechtkskim@hanmail.net (S.H.K.); taekmin82@gmail.com (T.M.N.); gebassist@naver.com (J.H.J.); unikkh@unitel.co.kr (K.H.K.); 2Department of Medicine, Sungkyunkwan University of School of Medicine, Suwon 16419, Republic of Korea; suhwonjun20@g.skku.edu; 3Division of Hematology and Medical Oncology, Department of Internal Medicine, Samsung Changwon Hospital, Sungkyunkwan University School of Medicine, Changwon 51353, Republic of Korea; tjrgus1@hanmail.net; 4Division of Neuro-Oncology and Department of Neurosurgery, Samsung Changwon Hospital, Sungkyunkwan University of School of Medicine, Changwon 51353, Republic of Korea

**Keywords:** brain metastasis, lung cancer, 5-aminolevulinic acid, genetic alteration, next-generation sequencing, infiltration, cell cycle regulation, proliferation

## Abstract

**Simple Summary:**

Although 5-aminolevulinic acid (ALA) is commonly used in glioma surgery to identify cell infiltration, recent studies have shown its successful use in brain metastasis (BM) surgery due to its ability to infiltrate adjacent brain tissue. Several studies have proven the histopathological relationship between tumor infiltration and positive fluorescence of 5-ALA; however, few comprehensive studies have shown the role of genetic alterations in the fluorescent activity of 5-ALA, especially in BM. The present study illustrates the causal relationship between certain genetic alterations (i.e., cell cycle regulation and cell proliferation) and the fluorescent activity of 5-ALA in BM. In addition, these alterations were associated with clinical outcomes of BM of lung adenocarcinoma. As the results were achieved through the clinical practice of BM surgery, such as intraoperative 5-ALA and next-generation sequencing (NGS) analysis, it is mandatory for basic researchers to examine the pathophysiology in more detail.

**Abstract:**

The primary objective of this study was to investigate the association of certain genetic alterations and intraoperative fluorescent activity of 5-aminolevulinic acid (ALA) in brain metastasis (BM) of lung adenocarcinoma. A retrospective cohort study was conducted among 72 patients who underwent surgical resection of BM of lung adenocarcinoma at our institute for five years. Cancer cell infiltration was estimated by the intraoperative fluorescent activity of 5-ALA, and genetic alterations were analyzed by next-generation sequencing (NGS). The sensitivity and specificity for detecting cancer cell infiltration using 5-ALA were 87.5% and 96.4%, respectively. Genes associated with cell cycle regulation (*p* = 0.003) and cell proliferation (*p* = 0.044) were significantly associated with positive fluorescence activity of 5-ALA in the adjacent brain tissue. Genetic alterations in cell cycle regulation and cell proliferation were also associated with shorter recurrence-free survival (*p* = 0.013 and *p* = 0.042, respectively) and overall survival (*p* = 0.026 and *p* = 0.042, respectively) in the multivariate analysis. The results suggest that genetic alterations in cell cycle regulation and cell proliferation are associated with positive fluorescence activity of 5-ALA in the adjacent infiltrative brain tissue and influence the clinical outcome of BM of lung adenocarcinoma.

## 1. Introduction

Brain metastasis (BM) has the top-ranked incidence of tumor of the central nervous system (CNS) in adults in the world, including the United States as well as Korea, whereby the ratio of BM and primary tumor of CNS is estimated at 5:1 [1,2]. BMs are reported to be found during treatment of systemic cancers or at the same time of systemic cancer diagnosis in about 8–10% patients [3,4,5]. In terms of the brain, common systemic cancers which frequently invade the brain are lung cancer, breast cancer, and melanoma. Melanoma has the strongest potency to invade the brain; as high as 40–60% patients with melanoma experience BM [6]. Among lung cancer, non-small cell lung cancer (NSCLC) has the highest frequency of BM, whereby 20% of patients are simultaneously diagnosed with BM and NSCLS [7], 10–20% of patients experience BM during treatment of NSCLC [8], and BM occurs in 40–50% of patients with stage III NSCLC [9]. Especially, anaplastic lymphocyte kinase (ALK) is a well-known receptor tyrosine kinase in NSCLC as well as in BM; BMs are found in 20–40% of patients with ALK-rearranged NSCLCs [9]. Despite an appropriate tyrosine kinase inhibitor being used for patients with ALK-rearranged NSCLCs, as many as 45–70% of patients suffer from BM [8]. As far as recent knowledge, the clinical outcomes of patients with BM are still dismal, where more than half of cancer patients cannot survive over 3–27 months after diagnosis of BM [3,4,5]. Recently, there has been a great advancement in systemic treatment for cancer patients including target therapies and immunotherapies. As a result, cancer patients can survive longer even they are in the advanced stage, which drives cancer patients to experience BM more frequently [5,10,11,12]. In considering the clinical importance of BMs, BMs usually make patients independent upon others, resulting in burdening the patients and their families with significant social and economic loads [13]. It is a largely different point from cancer metastasis to other organs except CNS that BM can shorten survival owing to the brain’s decreased performance status due to neurological dysfunctions, such as hemiparesis [14]. In fact, oncologists have many concerns in treating patients intensively due to these focal neurological deficits in clinical practice.

Among the several modern therapeutic options for BM, including surgery, radiation therapy, and systemic therapy options [15], the primary role of neurosurgery in the treatment of single BM is well established [16,17]. Recently, experts and practical guidelines recommend surgical resection in the following cases: (1) a limited number of BM, (2) large size of BM, (3) BM which is located in a safe area, (4) in case of necessity of tissue confirmation for pathological diagnosis, and (5) a huge mass producing neurological symptoms, for the improvement of such symptoms [16,17]. It is true that debulking surgery of BM can rapidly relieve mass effect and improve the neurological deterioration from increased intracranial pressure (ICP). Although steroid administration is considered as one of therapeutic strategies for treating ICP of BM, surgical removal can be more effective for refractory ICP symptoms and rapid reduction of ICP than intravenous steroid treatment. An additional benefit of the surgical resection of BM is to reduce the risk of prolonged treatment of steroids, dependency on steroids, and its potency of side effects from long-term usage and high dose administration. Recently, there have been revolutionary advancements in technology and concepts for the neurosurgical field [18], such as multimodal neuronavigation systems, awake surgery, intraoperative ultrasound, cortical mapping, sodium fluorescence, and 5-ALA (5-aminolevulinic acid) fluorescence. These newest technologies support maximal safe resection with fewer adverse effects and increase the potential for total resection. As a result, the clinical outcomes of patients who undergo the surgical resection of BM see more improvement [18,19,20]. Despite that BM is removed totally with assistance from high technologies, disastrous recurrence within the 2 cm boundary of the resection happens commonly, because the cancer cells have great capability to grow infiltratively, proliferate rapidly, and develop resistant clones against initial treatments shortly after treatment [21]. Therefore, there has been a report that the microscopically extensive resection of the adjacent infiltrative portion of BM can significantly reduce the recurrence rate of the resected site [22].

Intraoperative fluorescent activity of 5-ALA as one of the revolutionary techniques in brain tumor surgery is mainly used for glioma resection under fluorescence guidance [23]. 5-ALA is usually administered via the parenteral route and accumulates protoporphyrin IX in the tumor tissue, which shows active fluorescence in red color. This fluorescence of 5-ALA can be detected under the filtered light spectrum of a short wavelength. Recently, there have been several reports illustrating the active fluorescence of 5-ALA in BM resection [24,25]. Additionally, these reports suggest that active fluorescence of 5-ALA was found in the adjacent infiltrative margin beyond the tumor capsule after BM resection [24,25]. However, there are still controversies on the clinical application and significance of 5-ALA on BM resection because the fluorescent activity of 5-ALA has a heterogenous pattern even with positive findings in the majority of BM tissues [24,25]. Despite controversies, it is true that if 5-ALA fluorescence can illustrate the infiltration of the cancer cell into the adjacent tissue, then this method can be greatly helpful to remove BM completely as in glioma surgery.

Owing to advancements in genetic analysis in the field of oncology, genetic alterations have been included in the system to predict the prognosis of cancer patients. In the traditional prognostic system of brain metastasis, such as disease-specific Graded Prognostic Assessment (GPA), only clinical factors are included [26]. For example, the patient’s age at the time of BM diagnosis, Karnofsky Performance Scale (KPS) score, extracranial metastasis, and number of BM have been considered for the assessment of prognosis in BM patients with lung cancer [26]. Recently, genetic features have been used to assess the prognosis of patients with BM. Especially, for patients with BM of lung cancer, the genetic mutations of epithelial growth factor receptor (EGFR) and ALK, as well as known clinical factors, are considered to assess the prognosis of these patients [27]. Despite the popular application of genetic data for managing patients with BM, comprehensive studies have clarified the genetic role of BM biology, especially in the infiltration features of BM.

Herein, we primarily examined the intraoperative fluorescence activity of 5-ALA in the tumor resection cavity after the removal of BM of lung adenocarcinoma. Additionally, we examined the patterns of genetic alterations in BM samples obtained by surgical resection using next-generation sequencing (NGS). Finally, we determined the relationship between the fluorescence activity and genetic alterations using NGS. We also examined predictive factors associated with recurrence-free survival (RFS) and overall survival (OS) to validate previously known prognostic factors.

## 2. Materials and Methods

### 2.1. Patient Collection

This retrospective cohort study included patients who underwent surgical resection of BM of lung adenocarcinoma at our institute between March 2017 and June 2022. Lung adenocarcinoma was histopathologically confirmed at the time of BM diagnosis. We retrospectively reviewed medical records of patients with lung adenocarcinoma and BM. During this period, 235 patients were radiologically diagnosed with BM of lung adenocarcinoma at our institute. After establishing the diagnosis of BM of lung adenocarcinoma, our multidisciplinary team always collaborated to determine which option was the best treatment for individual patients. Among them, 72 underwent surgical resection of the BM. The inclusion criteria for surgical resection of BM were as follows: (1) patient’s life expectancy is longer than 3 months, (2) large lesion with edema and mass effect producing neurological symptoms, such as decline of mentation and hemiparesis, (3) lung adenocarcinoma is under control by systemic treatment, (4) patient has good performance status with active daily life, (5) the number of lesions is limited to three, and (6) instances of strong patient preference for surgical resection. Lesions that did not meet these criteria were treated with WBRT with simultaneous integrated boost (SIB) or SRS, palliative chemotherapy, or best supportive care.

### 2.2. Neurosurgical Resection and Application of 5-ALA

The principle of neurosurgical resection is en bloc removal without soiling cancer cells in the brain. The same neurosurgical technique was used because all surgical resections were performed by a single brain tumor surgeon (Y.Z.K.). After detecting the gap between the normal brain parenchyma and the pseudocapsule of the BM under a neurosurgical microscope, meticulous dissection was performed with protection from cancer cell soiling using a cottonoid barrier and no saline irrigation. The microscope was then switched to a fluorescent view of 5-ALA. The fluorescence pattern of the wall of the resection cavity was monitored and recorded. The fluorescence pattern was categorized as strong, vogue, or weak during the surgery. These categories were determined on the basis of decisions made by three individual neurosurgeons who participated in the operation, including the main operator and two assistant neurosurgeons. After surgery, the intraoperative 5-ALA findings were reviewed at a multidisciplinary conference. It was our policy for the application of 5-ALA in all neurosurgical resections of BM because of its capability to infiltrate the adjacent brain parenchyma. Nevertheless, the application of 5-ALA in the resection of BM was always determined by the operating neurosurgeon. Surgical extent was classified based on the BM’s capability of infiltration. Gross total resection (GTR) was defined as the simple removal of the BM en bloc, and microscopic complete resection (MCR) was defined as the additional removal of the portion with a positive fluorescent pattern adjacent to the BM capsule after GTR. However, we could not perform MCR in or near the eloquent area because of the risk of neurological sequelae. Additionally, more than two areas with strong fluorescence activity were obtained and sent to the pathological laboratory to detect cancer cell infiltration.

In principle, the patient takes the 5-ALA (Gliolan^®^, Photonamic GmbH & Co. KG, Pinneberg, Germany) at a dose of 20 mg/kg 2–4 h prior to anesthetic induction. A vial of Gliolan^®^ contains 1.5 g powder for oral solution and is reconstituted in 50 mL of drinking water (30 mg/mL) after opening. It takes approximately 1 h to prepare for the operation, including the anesthetic procedure, setting up the navigation system, and establishing intraoperative monitoring equipment, and another hour to encounter the tumor after skin incision. Therefore, it takes approximately 4–6 h after the administration of 5-ALA to determine the first fluorescent activity of 5-ALA in the superficial BM. As it takes an additional 1–3 h to complete the resection of the BM, the fluorescent activity of 5-ALA in the tumor cavity and tumor margin can be checked 5–9 h after 5-ALA administration. These steps of administration and application of 5-ALA were followed by manufactural protocol.

### 2.3. Clinical Assessment of Patients

The following clinical factors were examined retrospectively in the medical records: age, sex, KPS score, status of lung adenocarcinoma, the interval between the time of diagnosis of BM and lung adenocarcinoma, Recursive Partitioning Analysis (RPA) class at the time of diagnosis, and disease-specific GPA score. In this study, KPS scores were determined as described by Karnofsky et al. [28], whereby patients with a score of 70 or more were capable of caring for themselves, and those with a score of less than 70 required assistance to conduct activities of daily life. The RPA class was determined using the modified Radiation Therapy Oncology Group (RTOG) method [27], and the GPA score was assessed based on age, KPS, extracranial metastasis, and the number of BMs using the method of the largest data [26].

### 2.4. Radiological Features of Brain Metastasis and Lung Adenocarcinoma

All BMs were radiologically diagnosed by magnetic resonance image (MRI). The radiological features included the number of BM and the time interval between diagnosis of lung adenocarcinoma and BM. We examined the number of masses which had the gadolinium enhancement on T1 weighted MRI. In terms of the interval between the time of detecting BM in MRI and conforming lung adenocarcinoma histopathologically, BM which was diagnosed within 2 months from the time of diagnosis of lung adenocarcinoma was defined as synchronous and that diagnosed after 2 months from the time of diagnosis of lung adenocarcinoma was defined as metachronous. Furthermore, we examined extracranial metastasis using abdominal and chest computed tomography (CT). Simultaneously positron emission tomography (PET)-CT was performed at the time of BM diagnosis. The lung adenocarcinoma was classified as stable even in metachronous metastasis in the case of no interval change in the primary cancer on CT scan. But metachronous metastasis was classified as unstable in the case of growth of primary cancer or all synchronous metastasis cases.

Recurrence was defined as the presence of a new enhancing tumor mass at the resected site, as judged on the first postoperative MRI. New lesions at 2 cm or more out of the primary tumor resection cavity were classified as distant recurrence and not included in this analysis. Two individual neuroradiologists who did not have any clinical and pathological information evaluated the radiological features.

### 2.5. Genetic Alterations Using Next-Generation Sequencing

For the analysis of genetic alteration, ONCOaccuPanel^®^ (NGeneBio, Seoul, Republic of Korea) on the Illumina MiSeq platform was used for NGS. ONCOaccuPanel is a kind of hybridization capture-based DNA panel detecting somatic mutations and copy number alterations of 323 key cancer genes and fusions of 17 genes in solid tumors. ONCOaccuPanel DNA probes were designed for targeted sequencing of all exons and selected introns of 225 genes and partial exons of 98 genes (a total of 323 genes) (Appendix A). Formalin-fixed paraffin-embedded (FFPE) tissue specimens from 72 patients who underwent BM surgical resection were used for DNA extraction. Histological samples were obtained from the Archives of Pathology in our institute. The FFPE slices (5 µm thick) were deparaffinized and rehydrated with xylene and alcohol solutions. DNA was extracted and purified using a Maxwell FFPE Plus DNA Kit (Promega, Madison, WI, USA) according to the manufacturer’s instructions. The DNA quantity was determined by fluorometric quantification using a Quantus Fluorometer with a QuantiFluor dsDNA system (Promega). The DNA integrity number (DIN) was evaluated using an Agilent 4200 TapeStation (Agilent Technology, Santa Clara, CA, USA). Other processes for NGS analysis, such as library preparation and determination of coverage requirements and target region coverage, were performed as previously described [29].

The major functions of the genes were categorized based on 10 hallmarks of cancer [30]: tumor initiation (evading growth suppression), proliferation, apoptosis (resisting cell death, replicative immortality, and DNA repair after cell damage), angiogenesis, metabolism, epigenetic modification, destruction of immunity, invasion, and metastasis. Additionally, the main function of each gene was defined at the specialized website of the human gene database, The Gene Ontology Resource^®^ (www.geneontology.org: accessed on 30 March 2023) and GeneCards^®^ (www.genecards.org: accessed on 30 March 2023). On certain driver genes with multiple functions in cancer biology, a single role on major pathophysiology was engaged in the gene categorization.

### 2.6. Statistical Analysis for Recurrence-Free Survival and Overall Survival

We analyzed the medical records retrospectively of all patients who met the inclusion criteria to summarize the clinical course and radiographic results. Recurrence and date of death were examined precisely and recorded. RFS was defined as the time from the date of surgical resection to the date of detection of a new lesion in the resection cavity on MRI. Moreover, OS was defined as the time from the date of lung adenocarcinoma diagnosis to BM until death. The date of BM diagnosis was defined as the date of MRI scanning, whereas the date of lung adenocarcinoma diagnosis was defined as the date of biopsy or surgical resection of the lung lesion.

Statistical analyses were performed using SPSS ver. 20.0 (IBM Corp., Armonk, NY, USA). Differences between subgroups were analyzed using Student’s *t*-test for normally distributed continuous values, Mann–Whitney U test for non-normally distributed continuous values, and chi-squared tests to analyze categorical variables. RFS and OS were calculated using the Kaplan–Meier method. Comparisons between groups were performed using the log-rank test. Variables that were significantly associated with longer RFS and OS in patients with lung adenocarcinoma with BM in univariate analyses were examined using multivariate analysis. Several additional variables associated with RFS and OS in the literature and of interest to the investigators were also included in the multivariate analysis. In this analysis, the Cox proportional hazards regression model was used to assess the independent effects of specific factors on RFS and OS, and to define the hazard ratios of significant covariates. *p* values < 0.05 were considered statistically significant.

## 3. Results

### 3.1. Clinical Features of Patients

The clinical data of 75 patients who underwent surgical resection of BM of lung adenocarcinoma between March 2017 and June 2022 were included in this analysis. Among them, three patients were excluded because of insufficient NGS or radiologic data. Therefore, 72 patients (40 males, 32 females) were enrolled in this study (Table 1). The mean age of these patients at the time of BM diagnosis was 62.9 years (range 34.5–85.0 years). Forty-two patients (58.3%) had good performance status (KPS ≥ 70) and 30 patients (41.7%) had poor performance status (KPS < 70). Single BM was confirmed in 38 patients (52.8%), 22 patients (30.6%) had brain oligometastases, and the other four patients (16.6%) had four or more BMs (Table 1). Extracranial metastases to the contralateral lung, adrenal gland, or bone were present in 56 patients (77.8%). Forty-five patients with lung adenocarcinomas (62.5%) were controlled with systemic treatment (cytotoxic or immunotherapy). Twenty-three patients (31.9%) experienced BMs within two months of the diagnosis of lung adenocarcinoma, and 49 patients (60.1%) had BMs two months after the diagnosis of lung adenocarcinoma. Ten (13.9%) patients were classified as RPA class I, 44 (61.1%) as RPA class II, and 18 (25.0%) as RPA class III. Forty patients had a GPA score of 0–2.5 and 32 patients had a GPA score of 3.0–4.0. Forty-eight patients (66.7%) underwent GTR and 24 patients (33.3%) underwent MCR. Most patients (79.2%) received active adjuvant treatment after surgical resection of BM (Table 1).

In terms of recurrence, the patients with KPS < 70, those who underwent GTR, and those who received conservative treatment only after surgical resection of BMs had statistically higher recurrence rates than those with KPS ≥ 70, those who underwent MCR, and those who received active adjuvant treatment after surgical resection of BM (Table 1).

### 3.2. Genetic Alteration in Next-Generation Sequencing Analysis

In total, 49 gene alterations were detected in the BM using NGS (Appendix A). Genetic alterations were clustered according to the role of the gene (Figure 1). Eight genes associated with cell cycle regulation, namely, *CDKN2A*, *TP53*, *RB1*, *CDK4*, *CDK6*, *ATR*, *APOBEC3B*, and *LRP1B*, were altered in 53 (73.6%) BM samples. Seven genes associated with DNA repair, namely, *POLE*, *ATM*, *MLH1*, *BRCA2*, *MSH2*, *ZNF141*, and *ZNF563* were altered in 32 (44.4%) BM samples. Eleven genes associated with tumorigenesis, including *EPHA3*, *ALK*, *NOTCH1*, *RET*, *PTCH1*, *MET*, *SMO*, *STK11*, *ABL2*, *NF1*, and *APC*, were altered in 41 (56.9%) BM samples. Nineteen genes that are associated with proliferation, such as *mTOR*, *TERT*, *KRAS*, *PIK3CB*, *EGFR*, *PTEN*, *ERBB3*, *ERBB4*, *AKT3*, *MYC*, *NTRK1*, *RICTOR*, *PICTOR*, *HRAS*, *KIT*, *ARAF*, *SMAD4*, *KRT32*, and *KDR*, were altered in 48 (66.7%) BM samples. Three genes associated with epigenetic regulation, *ARID1A*, *KMT2A*, and *BRD3*, were altered in 22 (30.6%) BM samples. One gene associated with the destruction of the immune system, *EPPK1*, was altered in two (2.8%) BM samples (Table 2).

In terms of recurrence, genetic alteration in the genes that play a major role in cell cycle regulation was significantly associated with a high rate of recurrence (*p* = 0.008). Although there was no statistically significant association with recurrence, alterations in genes that play a major role in cellular proliferation tended to be associated with a high rate of recurrence (*p* = 0.088) (Table 2).

### 3.3. Intraoperative Fluorescence Patterns of 5-ALA

Among the 72 BM resections, 56 operations (77.8%) showed positive intraoperative fluorescent activity with a pink color. Histopathologically, there was no cancer cell infiltration in two samples (3.6%) which were obtained from the area with positive fluorescence activity, and 54 samples (96.4%) had cancer cell infiltration around the adjacent tissue with positive fluorescence activity. The two samples with no infiltration were tinged with blood. Sixteen operations (22.2%) yielded negative fluorescence. Fourteen samples (87.5%) did not show cancer cell infiltration histopathologically; however, cancer cell infiltration was found in two samples (12.5%) with negative fluorescence activity. The sensitivity was 87.5% and specificity was 96.4% for detecting cancer infiltration during the 5-ALA analysis.

Among the 53 BMs with alterations in genetic expression associated with cell cycle regulation, 41 BMs (77.4%) showed positive intraoperative fluorescent activity with pink color in the resection cavity (Table 3). Among the 32 BMs with alterations in gene expression that were associated with DNA repair, six BMs (18.8%) showed positive intraoperative fluorescent activity with a pink color in the resection cavity (Table 3). Among the 41 BMs with alterations in gene expression that were associated with tumorigenesis, 12 (29.3%) showed positive intraoperative fluorescent activity with pink color in the resection cavity (Table 3). Among the 48 BMs with alterations in gene expression that were associated with cancer proliferation, 31 (64.6%) showed positive intraoperative fluorescent activity with pink color in the resection cavity (Table 3). Among the 22 BMs with alterations in genetic expression that were associated with epigenetic regulation, two (9.1%) showed positive intraoperative fluorescent activity with pink color in the resection cavity (Table 3). Among the two BMs with alterations in genetic expression associated with epigenetic regulation, none showed positive intraoperative fluorescent activity with a pink color in the resection cavity (Table 3). These positive intraoperative fluorescent activities in the resection cavity were significantly associated with genetic alterations that play major roles in cell cycle regulation and cancer proliferation (Table 3).

### 3.4. Clinical Outcomes with Clinical Predisposing Factors

Mean follow-up duration was 12.4 months (ranging from 3.2 to 20.3 months). During follow-up, 28 patients (38.9%) experienced local recurrence at the BM resection site. Mean time to recurrence was 10.0 months (ranging from 4.6 to 14.0 months). Mean RFS was 14.6 months (95% confidence interval [CI], 13.8–15.4 months). Univariate analysis for predisposing factors of RFS in the BM of lung adenocarcinoma showed that the following clinical factors were associated with longer RFS in patients with KPS ≥ 70 than those with KPS < 70 (*p* = 0.009), in patients who underwent MCR than those who underwent GTR (*p* = 0.002), and in patients who received adjuvant radiation therapy and/or chemotherapy than those who received conservative treatment (*p* < 0.001) (Table 4). The Kaplan–Meier survival curve analysis showed the same results (Figure 2).

In terms of survival, the mean OS was 16.3 months (95% CI, 15.2–17.6 months). Forty-six patients (63.9%) succumbed to progression of lung adenocarcinoma. Univariate analysis for predisposing factors of OS showed that longer OS was observed in patients aged < 65 years than those aged ≥ 65 years (*p* = 0.038); patients with KPS ≥ 70 than those with KPS < 70 (*p* = 0.012); patients with stable lung adenocarcinoma than those with unstable disease (*p* = 0.019); patients with synchronous BM than those with metachronous BM (*p* = 0.042); patients with RPA class III than II or I (*p* = 0.026 and *p* < 0.001, respectively); patients with a GPA score 0–2.5 than 3.0–4.0 (*p* < 0.001); and patients who received adjuvant radiation therapy and/or chemotherapy than those who received conservative treatment (*p* < 0.001) (Table 5). The Kaplan–Meier survival curve analysis showed the same results (Figure 3).

### 3.5. Next-Generation Sequencing Data Predisposing Clinical Outcome

Patients with altered genes associated with cell cycle regulation had statistically shorter mean RFS than those without the altered genes (13.19 months vs. 18.88 months; *p* = 0.004). Despite having no statistically significant difference, patients with altered genes associated with cell proliferation had a tendency of shorter mean RFS than those without the altered genes (13.52 months vs. 16.78 months; *p* = 0.174) (Table 6). Other genetic alterations did not significantly influence RFS in patients with BM. The Kaplan–Meier survival curve analysis showed the same results (Figure 4).

The survival results were similar to those for RFS. Patients with altered genes associated with cell cycle regulation had statistically shorter mean OS than those without the altered genes (14.51 months vs. 20.20 months; *p* = 0.002), and patients with altered genes associated with cell proliferation had statistically shorter OS than those without the genes (15.19 months vs. 18.00 months; *p* = 0.001) (Table 7). The Kaplan–Meier survival curve analysis showed the same results (Figure 5).

### 3.6. Multivariate Analysis of Predicting Factors for Clinical Outcomes

In terms of RFS, several verified factors were independently associated with longer RFS in the literature, such as KPS ≥ 70 versus <70 (hazard ratio [HR] of 3.247; 95% CI 1.481–5.010), RPA class I versus III (HR of 2.913; 95% CI 1.205–4.621), MCR versus GTR (HR of 6.416 versus 8.415l; 95% CI 4.417–8.415), and active adjuvant treatment after surgical resection versus best supportive care (HR of 8.328; 95% CI 6.748–9.908). In addition, two unique alterations in genes, cell cycle regulation (HR of 3.568; 95% CI 1.709–5.427) and cell proliferation (HR of 2.992; 95% CI 1.488–4.496), were associated with longer RFS in the BM of lung adenocarcinoma (Table 8). However, the status of the primary cancer, which tended to be associated with RFS in univariate analysis, was not independently associated with RFS.

In the same way, multivariate analysis using Cox proportional hazards regression model showed that the following verified factors in the literature were independently associated with longer OS; age ≥ 65 years versus <65 years (HR of 2.315; 95% CI 1.284–3.346), KPS ≥ 70 versus <70 (HR of 3.138; 95% CI 2.024–4.452), stable primary cancer versus unstable primary cancer (HR of 2.887; 95% CI 1.865–3.909), RPA class I versus II (HR of 3.029; 95% CI 1.612–4.446), RPA class I versus III (HR of 6.534; 95% CI 4.325–8.743), RPA class II versus III (HR of 2.632; 95% CI 1.521–3.743), GPA score of 0–2.5 versus 3.0–4.0 (HR of 4.274; 95% CI 2.008–6.541), and active adjuvant treatment after surgical resection versus best supportive care (HR of 8.968; 95% CI 5.273–12.663). In addition, two unique alterations of genes that were independently associated with RFS, cell cycle regulation (HR of 3.816; 95% CI 1.947–5.685) and cell proliferation (HR of 2.681; 95% CI 1.543–3.819), were associated with longer OS in the BM of lung adenocarcinoma (Table 9). However, several factors that showed a tendency to be associated with OS in the univariate analysis, such as the time interval between BM and lung adenocarcinoma, the extent of BM resection, and alteration of tumorigenesis-associated genes, were not independently associated with OS.

## 4. Discussion

The present study shows that a high rate of BM had positive fluorescent activity of 5-ALA around the brain tissue even after en bloc resection, and these fluorescent activities were associated with specific types of genetic alterations, such as proliferation-associated and cell-cycle regulation-associated genes. As far as recent studies have been concerned, it is the first study showing the relationship between genetic alteration and BM infiltration which were estimated by 5-ALA and NGS analysis.

Several studies have reported the histopathological characteristics of areas with positive fluorescent activity of 5-ALA during BM resection [31,32]. Utsuki et al. [31] first documented the presence of 5-ALA-induced protoporphyrin IX in human metastatic brain tumors and found that protoporphyrin IX produced by tumor cells can leak into peritumoral tissues. They also reported that protoporphyrin IX fluorescence could be detected in peritumoral areas free of cancer cells. However, the authors suggested that this phenomenon could be explained by photobleaching. In fact, protoporphyrin IX is destroyed photochemically by light irradiation; in photobleaching, the fluorescence of 5-ALA diminishes very rapidly, and its elimination from the tissue occurs in proportion to the amount of protoporphyrin IX in the tissue [33,34]. It has been shown that a mass produces a larger amount of protoporphyrin IX than the tissue surrounding it, into which protoporphyrin IX leaks. This finding corresponds with the observation that protoporphyrin IX fluorescence is greater in the tumor than in the region in which it infiltrates. Therefore, to increase the accuracy of 5-ALA fluorescence in detecting infiltrative tissue around tumors, it is important to minimize the time of exposure to the light source of the neurosurgical microscope. In an effort to overcome the technical limitations of conventional light microscopic exposure of BM with positive 5-ALA fluorescence, a trial used endoscopy to visualize 5-ALA fluorescence at the margin of the resection cavity instead of light microscopy during surgery [35].

Mercea et al. [32] suggested an association between cancer cell infiltration and angiogenesis as estimated by intraoperative 5-ALA fluorescence activity. The hypothesis is based on the fact that the infiltrative behavior of BM consists either of growth along pre-existing blood vessels in a so-called “vascular co-option” growth pattern or a diffuse “glioma-like” single cell infiltration of peritumoral brain tissue [36]. They showed that angiogenesis was observed in 15% of the specimens from the peritumoral brain tissue and that angiogenesis was only found in the fluorescent brain samples. In contrast, angiogenesis was never detected in samples from non-fluorescent peritumoral brain tissues [32]. However, they failed to show a significant relationship between the 5-ALA fluorescence status of peritumoral brain tissue and tumor cell infiltration. Tumor cell infiltration was observed in the peritumoral brain tissue, with visible 5-ALA fluorescent and non-fluorescent activity. In contrast, our study showed a positive relationship between 5-ALA fluorescence activity in peritumoral brain tissue and tumor cell infiltration. The opposite result may be originated from the following reasons: (1) we obtained more than two samples in the peritumoral tissue (median: 4 and range: 1–6) for detecting infiltrative cancer cells, while they collected relatively small numbers of samples at peritumoral tissue (median: 1 and range: 1–4); (2) we counted “positive” samples with only strong fluorescence activity in pink color rather than vogue fluorescence, while they included the vogue fluorescence in “positive” samples. In general, their fluorescence effects have a vogue-like appearance. As mentioned above, the discrepancy in fluorescent activity may originate from the time of exposure to the light source in the tumor and peritumoral areas during surgery.

As our results suggest the infiltrative capability of cancer cells into brain tissue, the basic concept has changed. Until recently, there have been many reports of a decline in BM as circumscribed and non-infiltrating lesions [36,37,38,39]. The majority of BMs infiltrate the adjacent brain tissue, and this infiltration is correlated with a worse prognosis compared to BM without any evidence of infiltration [36]. Our previous study showed that complete microscopic resection, including adjacent infiltrative areas, could lower the local recurrence rate without adjuvant radiotherapy from 43.1% to 23.3% [22]. However, there are still disputing issues about the accuracy of 5-ALA for detecting infiltrative cancer cells in the brain tissue. For example, metastatic tissues appear to be highly heterogeneous and are usually highly vascularized. In addition, blood absorbs fluorescence and intraoperative impressions may vary. Additionally, there are instances where the interior of a tumor is lacking, while the surrounding brain tissue exhibits the fluorescence activity of 5-ALA [24]. Despite these disputing issues, it is acceptable that a high percentage of BMs, including adjacent infiltrative areas, are strongly positive for 5-ALA fluorescence activity during surgical resection of the BM with high sensitivity and specificity [24,31,39,40].

However, the present study has a limitation in terms of showing the direct effect of 5-ALA on planning the surgical extent as well as local control because the eloquence of the BM location is mainly a factor in planning the surgical extent, and local control is dependent on the surgical extent. Even in areas with strong fluorescent activity of 5-ALA, if they are eloquent, we cannot resect them further because of the risk of neurological morbidity. Therefore, there were certain portions of peritumoral tissue with a strong fluorescent activity that were extensively resected, while other portions with strong fluorescent activity were not extensively resected. Therefore, we cannot explain the direct effect of 5-ALA on local control of BM in terms of RFS. However, microscopically, complete resection should have a longer RFS than conventional gross total resection, as presented in this study.

To the best of our knowledge, few comprehensive studies have demonstrated the genetic and molecular characteristics of infiltrative BM according to the fluorescence activity of 5-ALA. Several studies have focused on the role of 5-ALA in glioblastoma [41,42,43,44]. For example, negative 5-ALA fluorescence has been reported to promote temozolomide resistance [41], and different patterns of immune infiltration have been found according to 5-ALA signatures by analyzing TCGA mRNA data [42,43]. Positive 5-ALA gene signatures, which were analyzed by spatially resolved bulk RNA profiling, showed transcriptionally concordant glioblastoma and myeloid cells with mesenchymal subtype to be associated with poor survival and recurrence of glioblastoma [44]. Although there is a report suggesting an association between cancer cell infiltration and angiogenesis, which was estimated by intraoperative 5-ALA fluorescence activity [32], their study was based on histopathological rather than genetic analysis. Since the role of intraoperative 5-ALA is to determine the infiltration of cancer cells, most published genetic studies have also focused on the association between the fluorescence of 5-ALA and the infiltrability of glioblastoma cells. Interestingly, the alteration of genes involved in cell cycle regulation was associated with the positive fluorescence activity of 5-ALA in cancer cell infiltration into the brain tissue, as shown in the present study. Among these, the alteration of *RB1* was the most common and was found in 38 samples (52.8%). Alterations in *CDKN2A* were the second most common and were found in 34 samples (47.2%). In addition, relatively high rates of the genetic alterations of *RB1* (63.2%) and *CDKN2A* (85.3%) were found among the samples with positive fluorescence activity of 5-ALA. However, the present study analyzed a relatively small number of genes included in the NGS panel (323 genes). Moreover, we did not analyze whole genes extensively, as Lang et al. [42] used TCGA mRNA data to determine the association between 5-ALA signatures and immune cell infiltration into glioblastoma. The ONCOaccuPanel^®^ used in this study did not include many genes associated with immuno-oncology, which is an emerging concern in cancer biology research. Despite these limitations, NGS panels can be a useful option in clinical practice, such as determining the association between genetic alterations and cancer cell infiltration into the brain tissue, rather than a comprehensive research area.

Although our study showed a meaningful relationship between the positive fluorescent activity of 5-ALA during surgical resection of BM of lung adenocarcinoma and certain genetic alterations, it had additional limitations. We identified two major concerns of this study: (1) there can be strong arguments in terms of methods of whether the fluorescence activity of 5-ALA can reflect the whole status of BM infiltration into the adjacent tissue, and (2) NGS analysis can illustrate whole genetic alterations in the part of BM infiltration. Moreover, it is not certain whether our assessment of the fluorescent activity of 5-ALA during surgical resection is always correct because the interpretation of the results obtained by photography may be subjective. Despite this, we simply classified the fluorescent activity of 5-ALA as “positive” or “negative,” and there was no clear cutoff value for determining the fluorescent activity of 5-ALA. To overcome this limitation, we always took photographs of the intraoperative findings and determined the intraoperative fluorescent activity with the agreement of attending neurosurgeons during the operation. In addition, we reviewed photographs of the intraoperative findings of 5-ALA in a multidisciplinary conference to validate the decision. It is necessary to develop the equipment to digitalize the strength of 5-ALA activity or quantify the fluorescence activity of 5-ALA. If it is possible to apply the equipment during the surgery, a clear cut for determining the 5-ALA fluorescence activity as positive and negative can be investigated after validation. In the near future, a deep learning-based method using artificial intelligence that can automatically determine the intraoperative fluorescent activity of 5-ALA in any selected area of the entire tumor resection cavity may be developed.

Finally, another limitation of this study was the bias originating from its retrospective design. This limitation could be overcome if the number of patients was sufficiently high. However, our study involved a small number of patients and may not have met the full assumptions of the statistical tests used. To reduce this bias, we obtained clinical data from computerized data archives using a uniform system and included candidate patients treated using the same protocol in a single center. The researchers involved in this study did not have any clinical information or experimental results to help avoid preconceptions. Pathological findings and radiological features were also independently reviewed; however, there was no clear bias due to the retrospective nature of the analysis. Despite these efforts, the conclusions drawn from our study require further validation through prospective and randomized clinical trials.

## 5. Conclusions

In the present study, we investigated the intraoperative fluorescence activity of 5-ALA in the tumor resection cavity after the removal of BM of lung adenocarcinoma. We found high sensitivity and specificity for 5-ALA in detecting cancer cell infiltration into the adjacent brain tissue. Additionally, using NGS analysis, we found that the alteration of genes associated with cell cycle regulation and cancer cell proliferation in BM was related to the positive fluorescent activity of 5-ALA. Although there can be strong arguments in terms of methods, whether the fluorescence activity of 5-ALA can reflect the whole status of BM infiltration into adjacent tissue, and whether NGS analysis can illustrate whole genetic alterations in the part of BM infiltration, these findings can be useful for researchers to drive further comprehensive studies to widen the scientific evidence for BM infiltration into the brain tissue.

## Figures and Tables

**Figure 1 cancers-16-00088-f001:**
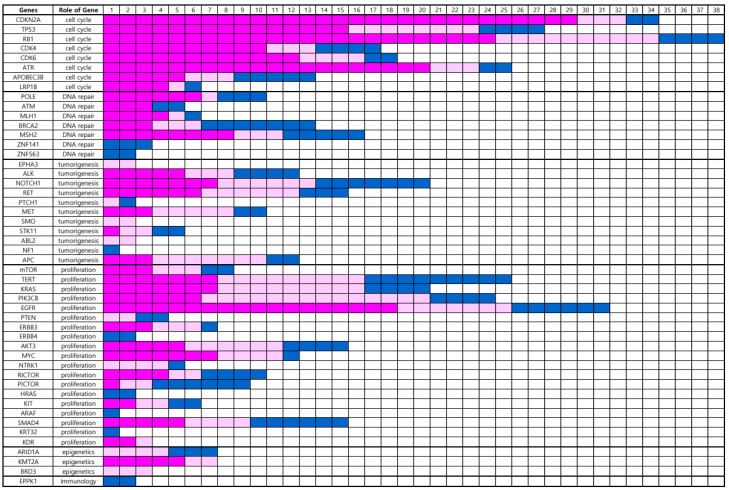
Genetic alterations according to the role of the genes by next-generation sequencing analysis; the blocks of bright pink color show strong positive fluorescent activity of 5-ALA, those of pale pink color shows vague fluorescent activity of 5-ALA, and those of blue color show negative fluorescent activity of 5-ALA.

**Figure 2 cancers-16-00088-f002:**
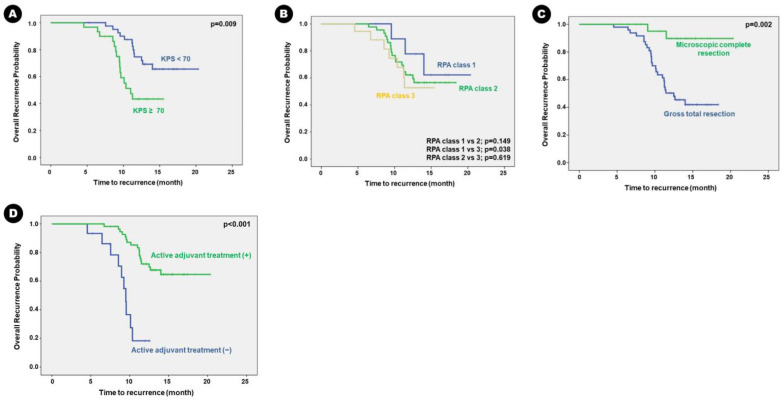
Kaplan–Meier survival curves for clinical predisposing factors of recurrence-free survival in the patients: (**A**) KPS, (**B**) RPA class, (**C**) extent of surgical resection, and (**D**) adjuvant treatment.

**Figure 3 cancers-16-00088-f003:**
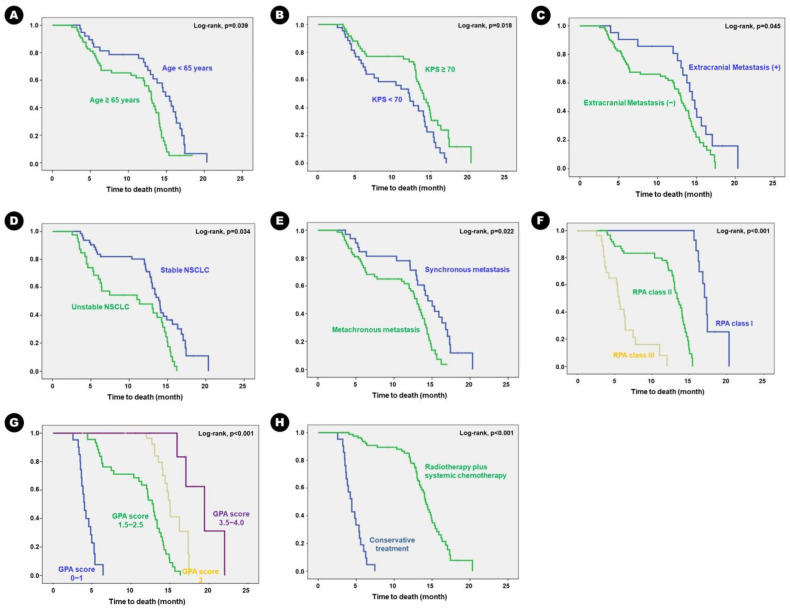
Kaplan–Meier survival curves for clinical predisposing factors of overall survival in the patients: (**A**) age, (**B**) KPS, (**C**) extracranial metastasis, (**D**) status of primary cancer, (**E**) timing of metastasis, (**F**) RPA class, (**G**) GPA score, and (**H**) adjuvant treatment.

**Figure 4 cancers-16-00088-f004:**
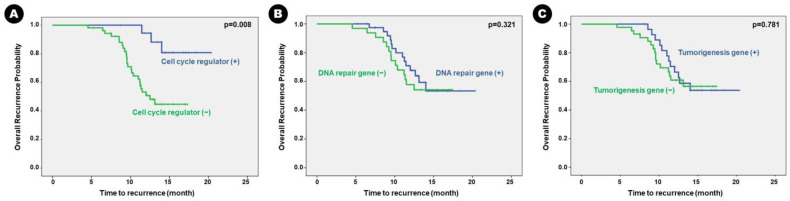
Kaplan–Meier survival curves for genetic alterations of recurrence-free survival in the patients. Genetic alteration of (**A**) cell-cycle regulation, (**B**) DNA repair, (**C**) tumorigenesis, (**D**) proliferation, (**E**) epigenetic mechanism, and (**F**) cancer immunology.

**Figure 5 cancers-16-00088-f005:**
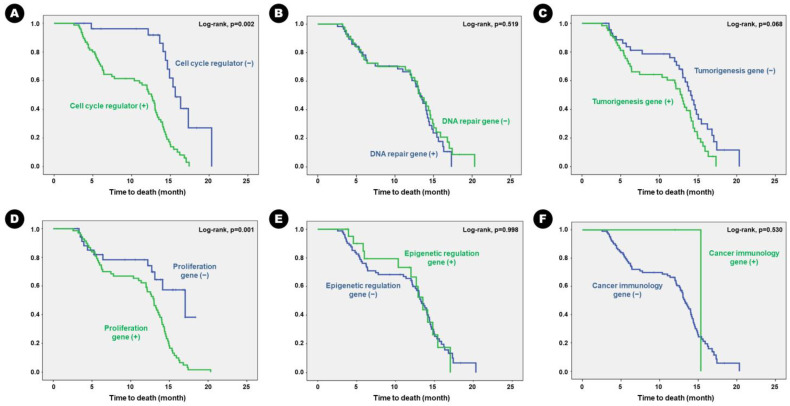
Kaplan–Meier survival curves for genetic alterations of overall survival in the patients. Genetic alteration of (**A**) cell-cycle regulation, (**B**) DNA repair, (**C**) tumorigenesis, (**D**) proliferation, (**E**) epigenetic mechanism, and (**F**) cancer immunology.

**Table 1 cancers-16-00088-t001:** Clinical characteristics of the patients with brain metastasis of lung adenocarcinoma who underwent the resection of brain metastasis.

	Total (n = 72)	Recurrence (n = 28)	No Recurrence (n = 44)	*p* Value
Mean age (years)	62.9 (34.5–85.0)	64.1 (34.5–81.2)	62.1 (54.3–85.0)	0.562
Male:Female	40:32	16:12	24:20	0.586
KPS < 70	30 (41.7%)	15 (53.6%)	15 (34.1%)	0.017
≥70	42 (58.3%)	13 (46.4%)	29 (65.9%)	
Number of brain metastasis				
Single	38 (52.8%)	13 (46.4%)	25 (56.8%)	0.651
Oligometastasis (2–3)	22 (30.6%)	10 (35.7%)	12 (27.3%)	
Multiple (>3)	12 (16.6%)	5 (17.9%)	7 (15.9%)	
Extracranial metastasis				
Yes	56 (77.8%)	22 (78.6%)	34 (77.3%)	0.873
No	16 (22.2%)	6 (21.4%)	4 (22.7%)	
Status of primary cancer				
Stable	45 (62.5%)	15 (53.6%)	30 (68.2%)	0.508
Unstable	27 (37.5%)	13 (46.4%)	14 (31.8%)	
Time interval of brain metastasis				
Synchronous (≤2 months)	23 (31.9%)	8 (28.6%)	15 (34.1%)	0.541
Metachronous (>2 months)	49 (60.1%)	20 (71.4%)	29 (65.9%)	
RPA class I	10 (13.9%)	3 (10.7%)	7 (15.9%)	0.829
II	44 (61.1%)	18 (64.3%)	26 (59.1%)	
III	18 (25.0%)	7 (25.0%)	11 (25.0%)	
GPA score 0–2.5	40 (55.6%)	16 (57.1%)	24 (54.5%)	0.913
3.0–4.0	32 (44.4%)	12 (42.9%)	20 (45.5%)	
Extent of resection GTR	48 (66.7%)	26 (92.9%)	22 (50.0%)	0.004
MCR	24 (33.3%)	2 (7.1%)	22 (50.0%)	
Adjuvant treatment after surgery				0.002
Conservative treatment *	15 (20.8%)	10 (35.7%)	5 (11.4%)	
RTx and/or CTx	57 (79.2%)	18 (64.3%)	39 (88.6%)	

Abbreviation. CTx, chemotherapy; GPA, graded prognostic assessment; GTR, gross total resection; KPS, Karnofsky performance scale; MCR, microscopic complete resection; RPA, recursive partitioning analysis; RTx, radiation therapy. * Conservative treatment included best supportive care without adjuvant therapy after surgery.

**Table 2 cancers-16-00088-t002:** Summaries of the next-generation sequencing (NGS) data from brain metastasis of lung adenocarcinoma.

Alteration of Genes	Total (n = 72)	Recurrence (n = 28)	No Recurrence (n = 44)	*p* Value
Genes associated with cell cycle regulation ^(1)^				
Yes	53 (73.6%)	25 (89.3%)	28 (63.6%)	0.008
No	19 (26.4%)	3 (11.7%)	16 (36.4%)	
Genes associated with DNA repair ^(2)^				
Yes	32 (44.4%)	14 (50.0%)	18 (40.9%)	0.217
No	40 (65.6%)	14 (50.0%)	26 (59.1%)	
Genes associated with tumorigenesis ^(3)^				
Yes	41 (56.9%)	16 (57.1%)	25 (56.8%)	0.904
No	31 (43.1%)	12 (42.9%)	19 (43.2%)	
Genes associated with proliferation ^(4)^				
Yes	48 (66.7%)	20 (71.4%)	28 (63.6%)	0.088
No	24 (33.3%)	8 (28.6%)	16 (36.4%)	
Genes associated with epigenetic regulation ^(5)^				
Yes	22 (30.6%)	9 (32.1%)	13 (29.5%)	0.887
No	50 (69.4%)	19 (61.9%)	31 (70.5%)	
Genes associated with cancer immunology ^(6)^				0.962
Yes	2 (2.8%)	1 (3.6%)	1 (2.3%)	
No	70 (97.2%)	27 (96.4%)	43 (97.7%)	

(1) *CDKN2A*, *TP53*, *RB1*, *CDK4*, *CDK6*, *ATR*, *APOBEC3B*, *LRP1B*; (2) *POLE*, *ATM*, *MLH1*, *BRCA2*, *MSH2*, *ZNF141*, *ZNF563*; (3) *EPHA3*, *ALK*, *NOTCH1*, *RET*, *PTCH1*, *MET*, *SMO*, *STK11*, *ABL2*, *NF1*, *APC*; (4) *mTOR*, *TERT*, *KRAS*, *PIK3CB*, *EGFR*, *PTEN*, *ERBB3*, *ERBB4*, *AKT3*, *MYC*, *NTRK1*, *RICTOR*, *PICTOR*, *HRAS*, *KIT*, *ARAF*, *SMAD4*, *KRT32*, *KDR*, (5) *ARID1A*, *KMT2A*, *BRD3*; (6) *EPPK1*.

**Table 3 cancers-16-00088-t003:** Association of the next-generation sequencing (NGS) data and intraoperative fluorescence activity of 5-ALA in the brain metastasis of lung adenocarcinoma.

Alteration of Genes	Total (n = 72)	Fluorescence Activity		*p* Value
		Strong	Vogue or Absence	
Genes associated with cell cycle regulation ^(1)^		45	27	
Yes	53 (73.6%)	41 (91.1%)	12 (44.4%)	0.003
No	19 (26.4%)	4 (8.9%)	15 (55.6%)	
Genes associated with DNA repair ^(2)^		13	59	
Yes	32 (44.4%)	6 (46.2%)	16 (27.1%)	0.175
No	40 (65.6%)	7 (53.8%)	33 (62.9%)	
Genes associated with tumorigenesis ^(3)^		21	51	
Yes	41 (56.9%)	12 (57.1%)	29 (56.8%)	0.922
No	31 (43.1%)	9 (42.9%)	22 (43.2%)	
Genes associated with proliferation ^(4)^		41	31	
Yes	48 (66.7%)	31 (75.6%)	17 (54.8%)	0.044
No	24 (33.3%)	10 (24.4%)	14 (45.2%)	
Genes associated with epigenetic regulation ^(5)^		5	67	
Yes	22 (30.6%)	2 (40.0%)	20 (29.8%)	0.797
No	50 (69.4%)	3 (60.0%)	47 (70.2%)	
Genes associated with cancer immunology ^(6)^		0	72	0.951
Yes	2 (2.8%)	0 (0.0%)	2 (2.3%)	
No	70 (97.2%)	0 (0.0%)	70 (97.7%)	

(1) *CDKN2A*, *TP53*, *RB1*, *CDK4*, *CDK6*, *ATR*, *APOBEC3B*, *LRP1B*; (2) *POLE*, *ATM*, *MLH1*, *BRCA2*, *MSH2*, *ZNF141*, *ZNF563*; (3) *EPHA3*, *ALK*, *NOTCH1*, *RET*, *PTCH1*, *MET*, *SMO*, *STK11*, *ABL2*, *NF1*, *APC*; (4) *mTOR*, *TERT*, *KRAS*, *PIK3CB*, *EGFR*, *PTEN*, *ERBB3*, *ERBB4*, *AKT3*, *MYC*, *NTRK1*, *RICTOR*, *PICTOR*, *HRAS*, *KIT*, *ARAF*, *SMAD4*, *KRT32*, *KDR*, (5) *ARID1A*, *KMT2A*, *BRD3*; (6) *EPPK1*.

**Table 4 cancers-16-00088-t004:** Univariate analysis for clinical predisposing factors of recurrent-free survival in the patients.

Clinical Factors	Mean RFS (95% CI)	Hazard Ratio (95% CI)	*p* Value
Age (years) ≥65	15.09 (13.35–16.83)		
<65	15.45 (13.82–17.11)	1.256 (0.754–1.758)	0.566
Gender Male	14.52 (13.06–15.99)		
Female	16.02 (14.12–17.91)	1.096 (0.888–1.304)	0.757
KPS <70	11.83 (10.53–13.13)		
≥70	17.11 (15.66–18.56)	6.820 (4.047–9.593)	0.009
Extracranial metastasis Yes	14.06 (12.19–15.93)		
No	15.73 (14.25–17.21)	1.009 (0.556–1.462)	0.923
Status of primary cancer Unstable	13.51 (11.63–14.66)		
Stable	16.51 (14.93–18.08)	1.937 (0.915–2.959)	0.164
Time interval of brain metastasis			
Metachronous (>2 months)	15.34 (13.70–16.98)		
Synchronous (≤2 months)	15.42 (13.75–17.08)	1.707 (0.795–2.619)	0.400
RPA class III	12.39 (10.65–14.14)		
II	14.71 (13.40–16.02)		
I	17.16 (14.24–20.07)	1.959 (0.911–3.007)	0.149
GPA score 0–2.5	14.13 (12.87–15.39)		
3.0–4.0	15.81 (13.83–17.79)	1.018 (0.617–1.419)	0.894
Extent of resection GTR	13.46 (12.16–14.76)		
MCR	19.30 (17.94–20.66)	10.602 (7.361–13.843)	0.002
Adjuvant treatment of brain metastasis			
Conservative treatment *	9.35 (8.21–10.49)		
RTx and/or CTx	16.93 (15.64–18.22)	13.488 (9.284–17.692)	<0.001

Abbreviation. CI, confidence interval; CTx, chemotherapy; GPA, graded prognostic assessment; GTR, gross total resection; KPS, Karnofsky performance scale; MCR, microscopic complete resection; RFS, recurrence-free survival; RPA, recursive partitioning analysis; RTx, radiation therapy. * Conservative treatment included best supportive care without adjuvant therapy after surgery.

**Table 5 cancers-16-00088-t005:** Univariate analysis for clinical predisposing factors of overall survival in the patients.

Clinical Factors	Mean OS (95% CI)	Hazard Ratio (95% CI)	*p* Value
Age (years) ≥65	14.47 (12.56–16.38)	1.00	
<65	20.07 (16.25–22.53)	2.21 (1.13–3.29)	0.038
Gender Male	16.21 (13.98–18.44)	1.00	
Female	16.45 (14.36–19.56)	1.27 (0.69–1.85)	0.805
KPS <70	13.32 (11.68–14.96)	1.00	
≥70	18.29 (15.71–22.07)	4.36 (2.52–6.21)	0.012
Number of brain metastasis			
Multiple	15.02 (13.82–17.22)	1.00	
Single + Oligometastasis	17.29 (15.05–20.85)	1.84 (0.83–2.85)	0.346
Extracranial metastasis Yes	16.11 (15.14–17.08)	1.00	
No	16.59 (15.37–17.09)	1.32 (0.88–1.76)	0.682
Status of primary cancer Unstable	13.76 (11.21–15.31)	1.00	
Stable	20.19 (16.19–24.35)	3.51 (1.74–5.28)	0.019
Time interval of brain metastasis			
Metachronous (>2months)	14.33 (12.28–16.38)	1.00	
Synchronous (≤2months)	20.31 (15.98–25.44)	2.46 (1.29–3.63)	0.042
RPA class III	11.76 (9.49–12.03)	1.00	
II	16.42 (13.64–19.21)	3.08 (1.54–4.62)	0.026
I	25.43 (21.62–29.24)	5.37 (2.80–7.94)	<0.001
GPA score 0–2.5	11.92 (9.12–13.72)	1.00	
3.0–4.0	24.09 (21.62–27.14)	7.58 (4.63–10.53)	<0.001
Extent of resection GTR	15.18 (13.05–17.72)	1.00	
MCR	18.27 (16.57–20.44)	2.05 (0.92–3.18)	0.067
Adjuvant treatment after surgery			
Conservative treatment *	8.76 (7.12–9.41)	1.00	
RTx and/or CTx	17.39 (15.26–19.42)	13.43 (6.71–20.05)	<0.001

Abbreviation. CI, confidence interval; CTx, chemotherapy; GPA, graded prognostic assessment; GTR, gross total resection; KPS, Karnofsky performance scale; MCR, microscopic complete resection; OS, overall survival; RPA, recursive partitioning analysis; RTx, radiation therapy. * Conservative treatment included best supportive care without adjuvant therapy after surgery.

**Table 6 cancers-16-00088-t006:** Univariate analysis for the next-generation sequencing (NGS) data predisposing recurrence-free survival in the patients.

Role of Genes	Mean RFS (95% CI)	Hazard Ratio (95% CI)	*p* Value
Cell cycle regulation ^(1)^			
Yes	13.19 (12.04–14.34)		
No	18.88 (17.41–20.36)	8.153 (5.741–10.565)	0.004
DNA repair ^(2)^			
Yes	13.98 (12.28–15.25)		
No	15.98 (14.23–17.73)	1.301 (0.894–1.708)	0.321
Tumorigenesis ^(3)^			
Yes	13.97 (12.64–14.44)		
No	16.07 (14.26–17.88)	1.177 (0.725–1.629)	0.781
Proliferation ^(4)^			
Yes	13.52 (12.34–14.70)		
No	16.78 (14.84–18.73)	2.197 (0.908–3.486)	0.174
Epigenetic regulation ^(5)^			
Yes	13.50 (11.52–15.47)		
No	16.05 (14.58–17.52)	1.472 (0.836–2.108)	0.492
Cancer immunology ^(6)^			
Yes	12.00 (7.91–16.08)		
No	15.77 (14.45–17.09)	1.192 (0.584–1.801)	0.737

Abbreviation. CI, confidence interval; RFS, recurrence-free survival. (1) *CDKN2A*, *TP53*, *RB1*, *CDK4*, *CDK6*, *ATR*, *APOBEC3B*, *LRP1B*; (2) *POLE*, *ATM*, *MLH1*, *BRCA2*, *MSH2*, *ZNF141*, *ZNF563*; (3) *EPHA3*, *ALK*, *NOTCH1*, *RET*, *PTCH1*, *MET*, *SMO*, *STK11*, *ABL2*, *NF1*, *APC*; (4) *mTOR*, *TERT*, *KRAS*, *PIK3CB*, *EGFR*, *PTEN*, *ERBB3*, *ERBB4*, *AKT3*, *MYC*, *NTRK1*, *RICTOR*, *PICTOR*, *HRAS*, *KIT*, *ARAF*, *SMAD4*, *KRT32*, *KDR*; (5) *ARID1A*, *KMT2A*, *BRD3*; (6) *EPPK1*.

**Table 7 cancers-16-00088-t007:** Univariate analysis for the next-generation sequencing (NGS) data predisposing overall survival in the patients.

Role of Genes	Mean OS (95% CI)	Hazard Ratio (95% CI)	*p* Value
Cell cycle regulation ^(1)^			
Yes	14.51 (13.41–15.60)		
No	20.20 (18.55–21.86)	10.896 (8.457–13.335)	0.002
DNA repair ^(2)^			
Yes	15.62 (14.36–16.89)		
No	16.14 (14.55–17.73)	1.417 (0.713–2.101)	0.519
Tumorigenesis ^(3)^			
Yes	14.92 (13.65–16.18)		
No	17.18 (15.61–18.75)	2.815 (0.978–4.652)	0.068
Proliferation ^(4)^			
Yes	15.19 (14.07–16.31)		
No	18.00 (16.01–19.99)	6.407 (4.216–8.598)	0.001
Epigenetic regulation ^(5)^			
Yes	15.80 (14.62–16.97)		
No	16.33 (14.37–18.30)	1.008 (0.465–1.551)	0.998
Cancer immunology ^(6)^			
Yes	19.34 (19.34–19.34)		
No	15.85 (14.81–16.88)	0.468 (0.311–0.625)	0.530

Abbreviation. CI, confidence interval; OS, overall survival. (1) *CDKN2A*, *TP53*, *RB1*, *CDK4*, *CDK6*, *ATR*, *APOBEC3B*, *LRP1B*; (2) *POLE*, *ATM*, *MLH1*, *BRCA2*, *MSH2*, *ZNF141*, *ZNF563*; (3) *EPHA3*, *ALK*, *NOTCH1*, *RET*, *PTCH1*, *MET*, *SMO*, *STK11*, *ABL2*, *NF1*, *APC*; (4) *mTOR*, *TERT*, *KRAS*, *PIK3CB*, *EGFR*, *PTEN*, *ERBB3*, *ERBB4*, *AKT3*, *MYC*, *NTRK1*, *RICTOR*, *PICTOR*, *HRAS*, *KIT*, *ARAF*, *SMAD4*, *KRT32*, *KDR*; (5) *ARID1A*, *KMT2A*, *BRD3*; (6) *EPPK1*.

**Table 8 cancers-16-00088-t008:** Multivariate analysis for predisposing factors of recurrence-free survival in the patients using the Cox regression model.

	Hazard Ratio (95% CI)	*p* Value
KPS (≥70 vs. <70)	3.247 (1.481–5.010)	0.020
Status of primary cancer (stable vs. unstable)	1.572 (0.953–2.191)	0.121
RPA class (I vs. II)	1.294 (0.643–1.945)	0.761
(I vs. III)	2.913 (1.205–4.621)	0.041
(II vs. III)	1.162 (0.651–1.673)	0.883
Extent of resection (MCR vs. GTR)	6.416 (4.417–8.415)	0.002
Active adjuvant treatment (Yes vs. No)	8.328 (6.748–9.908)	<0.001
Alteration of cell cycle regulatory gene (absence vs. presence)	3.568 (1.709–5.427)	0.013
Alteration of proliferation-associated gene (absence vs. presence)	2.992 (1.488–4.496)	0.042

Abbreviation. CI, confidence interval; GTR, gross total resection; KPS, Karnofsky performance scale; MCR, microscopic complete resection; RPA, recursive partitioning analysis.

**Table 9 cancers-16-00088-t009:** Multivariate analysis for predisposing factors of overall survival in the patients using the Cox regression model.

	Hazard Ratio (95% CI)	*p* Value
Age (≥65 years vs. <65 years)	2.315 (1.284–3.346)	0.046
KPS (≥70 vs. <70)	3.138 (2.024–4.452)	0.029
Status of primary cancer (stable vs. unstable)	2.887 (1.865–3.909)	0.034
Time interval of brain metastasis (synchronous vs. metachronous)	1.716 (0.884–2.548)	0.073
RPA class (I vs. II)	3.029 (1.612–4.446)	0.026
(I vs. III)	6.534 (4.325–8.743)	<0.001
(II vs. III)	2.632 (1.521–3.743)	0.041
GPA score (0–2.5 vs. 3.0–4.0)	4.274 (2.008–6.541)	0.008
Extent of resection (MCR vs GTR)	1.338 (0.726–1.949)	0.334
Active adjuvant treatment (Yes vs. No)	8.968 (5.273–12.663)	<0.001
Alteration of cell cycle regulatory gene (absence vs. presence)	3.816 (1.947–5.685)	0.026
Alteration of tumorigenesis-associated gene (absence vs. presence)	2.037 (0.938–3.136)	0.062
Alteration of proliferation-associated gene (absence vs. presence)	2.681 (1.543–3.819)	0.042

Abbreviation. CI, confidence interval; GPA, graded prognostic assessment; GTR, gross total resection; KPS, Karnofsky performance scale; MCR, microscopic complete resection; OS, overall survival; RPA, recursive partitioning analysis.

## Data Availability

The data presented in this study are available upon request from the corresponding author. These data are not publicly available due to privacy restrictions since they contain information that could compromise the privacy of the study participants.

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
