# Peer review of "Clinical Application of the Association between Genetic Alteration and Intraoperative Fluorescence Activity of 5-Aminolevulinic Acid during the Resection of Brain Metastasis of Lung Adenocarcinoma"

_cancers, 2023, doi:10.3390/cancers16010088_

Round 1

Reviewer 1 Report

Comments and Suggestions for Authors

The article mainly studies the relationship between 5-ALA guided fluorescence resection of lung adenocarcinoma brain metastasis and genetic changes. The following areas still need improvement.

1. The introduction should focus on the brain metastasis of lung adenocarcinoma and 5-ALA.

2. The cited references are too old and need to be updated, for example, "Wang, B.; Guo, H.; Xu, H.; Yu, H.; Chen, Y.; Zhao, G. Research Progress and Challenges in the Treatment of Central Nervous System Metastasis of Non-Small Cell Lung Cancer. Cells 202110, 2620. https://doi.org/10.3390/cells10102620"

3. The role of genes isn't presented in Supplementary Figure 1, and the role of genes in Figure 1 is different from your description in lines 231-237. You should unify the expression of these parts.

4. The image quality in the article is poor, please make improvements.

Author Response

December 14, 2023

AUTHOR’S REVISION LETTER

  • Ms. No. cancers-2780177
  • Ms. Title: Clinical application of the association between genetic alteration and intraoperative fluorescence activity of 5-aminolevulinic acid during the resection of brain metastasis of lung adenocarcinoma

Dear Reviewer,

I really thank the editor and reviewers of “cancers” for taking their time to review my manuscript in detail. Herein, I am going to submit the revised manuscript. I hope you find that the revised manuscript will better meet the requirements of “cancers”. We are willing to correct the problems again after review of the revised manuscript. I fully appreciated again the reviewer for their considerate and constructive reviews.

*The revised part according to the comment of was written in red color on manuscript.

Reviewer 1.

The article mainly studies the relationship between 5-ALA guided fluorescence resection of lung adenocarcinoma brain metastasis and genetic changes. The following areas still need improvement.

  1. The introduction should focus on the brain metastasis of lung adenocarcinoma and 5-ALA.

Author response) I appreciate your kind and detail comment. We deleted the paragraph describing general concept of treating brain metastasis, and focused on fluorescence activity of 5-ALA during surgical resection and its relationship with genetic alterations of brain metastasis in the section of Introduction.

  1. The cited references are too old and need to be updated, for example, "Wang, B.; Guo, H.; Xu, H.; Yu, H.; Chen, Y.; Zhao, G. Research Progress and Challenges in the Treatment of Central Nervous System Metastasis of Non-Small Cell Lung Cancer. Cells 2021, 10, 2620. https://doi.org/10.3390/cells10102620"

Author response) I appreciate your kind and detail comment. Actually, it is fact that so many papers have been published to the dates about brain metastasis of lung cancer. As reviewer’s comment, we cited the updated article in the manuscript.

  1. The role of genes isn't presented in Supplementary Figure 1, and the role of genes in Figure 1 is different from your description in lines 231-237. You should unify the expression of these parts.

Author response) I appreciate your kind and detail comment. The genes which are included in Supplementary Figure 1 and Figure 1 are same. The difference between 2 figures is the way of grouping. Supplementary Figure 1 shows that all the genetic alterations in the order of  expression frequency without regarding genetic function, and Figure 1 shows the cluster focused on genetic function of same genes in Supplementary Figure 1. As 2 figures included same genes, first figure was separated from main body and moved to supplementary file. As reviewer’s comment, we have changed Supplementary Figure 1 to express the role of function of individual gene. In fact, we as authors want readers to focus their attention to Figure 1 rather than Supplementary Figure 1, because certain genetic roles in BM biology are suggested to make association with fluorescence activity of 5-ALA.

  1. The image quality in the article is poor, please make improvements.

Author response) I appreciate your kind and detail comment. As reviewer commented, the quality of images is relatively poor, because we made images directly from conversion PPT file into JPGE file. So, we will receive the technical support from out institute for improving image quality. However, the due period for revision is much short as 5 days to complete the technical support. Therefore, we will provide the updated image during the process for publication.

Young Zoon Kim, M.D., Ph.D.

Director, Division of Neuro Oncology / Professor, Department of Neurosurgery,

Sungkyunkwan University School Medicine, Samsung Changwon Hospital.

Reviewer 2 Report

Comments and Suggestions for Authors

The authors present a nice retrospective review investigating the correlation between extratumoral 5-ALA spread and genetic variants within the brain metastases related to cell cycling and proliferation. This is an interest report and the data is presented in a sound way. My only concerns were address in the limitations section, namely the method of qualifying extratumoral spread. Is there a way to quantify the fluorescence to make the data acquisition more sound? Additionally, were there any tumors that could not be totally resected? What were their DNA profiles. Finally, a more in depth discussion of the decision making by the surgeon once fluorescence was found beyond the tumor would be helpful in order to discuss the options available in that scenario. If these can be address, I would feel more sound to recommend this manuscript for publication. 

Author Response

December 14, 2023

AUTHOR’S REVISION LETTER

  • Ms. No. cancers-2780177
  • Ms. Title: Clinical application of the association between genetic alteration and intraoperative fluorescence activity of 5-aminolevulinic acid during the resection of brain metastasis of lung adenocarcinoma

Dear Reviewer,

I really thank the editor and reviewers of “cancers” for taking their time to review my manuscript in detail. Herein, I am going to submit the revised manuscript. I hope you find that the revised manuscript will better meet the requirements of “cancers”. We are willing to correct the problems again after review of the revised manuscript. I fully appreciated again the reviewer for their considerate and constructive reviews.

*The revised part according to the comment of was written in blue color on manuscript.

Reviewer 2.

The authors present a nice retrospective review investigating the correlation between extra tumoral 5-ALA spread and genetic variants within the brain metastases related to cell cycling and proliferation. This is an interest report and the data is presented in a sound way. My only concerns were address in the limitations section, namely the method of qualifying extra tumoral spread. Is there a way to quantify the fluorescence to make the data acquisition more sound? Additionally, were there any tumors that could not be totally resected? What were their DNA profiles. Finally, a more in-depth discussion of the decision making by the surgeon once fluorescence was found beyond the tumor would be helpful in order to discuss the options available in that scenario. If these can be address, I would feel sounder to recommend this manuscript for publication.

Author response) I appreciate your comprehensive and detail comment.

1) I totally agree with reviewer’s opinion that there is no definite tool to quantify the fluorescence activity to make data acquisition. In fact, there were several times that we also had a difficulty to determine the strength of 5-ALA activity during the surgery. As described at the section of discussion (line 566-569), we reviewed photographs of the intraoperative findings of 5-ALA in a multidisciplinary conference to validate the decision for determining the strength of 5-ALA activity. “It is necessary to develop the equipment to digitalize the strength of 5-ALA activity or quantify the fluorescence activity of 5-ALA. If it is possible to apply the equipment during the surgery, clear cut for determining the 5-ALA fluorescence activity as positive and negative can be investigated after validation”. This sentence was added in the section of Discussion.

2) As described in the section of Materials and Method (Line 152-154), we tried to remove brain metastasis in en bloc fashion. Neurosurgeon who operated the surgical resection was only one and the operator (Y.Z.K) had over 20-years’ experience of brain tumor resection. Therefore, the way of surgical resection was uniform. All tumors were resected totally in en bloc fashion. So, there was no tumor which was not totally removed. But there was a difference in extent of resection, such as additional removal of adjuvant brain tissue or not (gross total resection [GTR] vs. microscopic complete resection [MCR]). There was no difference of genetic alteration between GTR group and MCR group, because the decision of GTR or MCR during the surgical resection was based on the eloquence of brain metastasis (Line 167-171). As above sentences were already described in the manuscript, we did not add more explanation in the manuscript.

Young Zoon Kim, M.D., Ph.D.

Director, Division of Neuro Oncology / Professor, Department of Neurosurgery,

Sungkyunkwan University School Medicine, Samsung Changwon Hospital.